# Recent Developments in Methicillin-Resistant *Staphylococcus aureus* (MRSA) Treatment: A Review

**DOI:** 10.3390/antibiotics11050606

**Published:** 2022-04-29

**Authors:** Palanichamy Nandhini, Pradeep Kumar, Suresh Mickymaray, Abdulaziz S. Alothaim, Jayaprakash Somasundaram, Mariappan Rajan

**Affiliations:** 1Biomaterials in Medicinal Chemistry Laboratory, Department of Natural Products Chemistry, School of Chemistry, Madurai Kamaraj University, Madurai 625021, India; nandhinipalanichamy4@gmail.com; 2Department of Pharmacy and Pharmacology, School of Therapeutic Sciences, Faculty of Health Sciences, University of the Witwatersrand, Johannesburg 2193, South Africa; pradeep.kumar@wits.ac.za; 3Department of Biology, College of Science, Al-Zulfi, Majmaah University, Majmaah 11952, Saudi Arabia; s.maray@mu.edu.sa (S.M.); a.alothaim@mu.edu.sa (A.S.A.); 4Department of Pharmaceutics, College of Pharmacy, University of Hafr Al Batin, Hafar Al Batin 39524, Saudi Arabia; sjayaprakash@uhb.edu.sa

**Keywords:** methicillin-resistant *Staphylococcus aureus* (MRSA), superbug, pathogenesis, antibiotics, treatment

## Abstract

*Staphylococcus aureus* (*S. aureus*) is a Gram-positive bacterium that may cause life-threatening diseases and some minor infections in living organisms. However, it shows notorious effects when it becomes resistant to antibiotics. Strain variants of bacteria, viruses, fungi, and parasites that have become resistant to existing multiple antimicrobials are termed as superbugs. Methicillin is a semisynthetic antibiotic drug that was used to inhibit staphylococci pathogens. The *S. aureus* resistant to methicillin is known as methicillin-resistant *Staphylococcus aureus* (MRSA), which became a superbug due to its defiant activity against the antibiotics and medications most commonly used to treat major and minor infections. Successful MRSA infection management involves rapid identification of the infected site, culture and susceptibility tests, evidence-based treatment, and appropriate preventive protocols. This review describes the clinical management of MRSA pathogenesis, recent developments in rapid diagnosis, and antimicrobial treatment choices for MRSA.

## 1. Introduction

### 1.1. Evolution of MRSA Treatment

*Staphylococcus aureus* (*S. aureus*) is a Gram-positive, clustered, spherical-shaped bacterium; it is primarily a human and animal pathogen. It belongs to the family of Staphylococcaceae and the genus known as Staphylococcus. A Scottish surgeon, Sir Alexander Ogston (1881), while performing a procedure, discovered that Staphylococcus could cause wound infections in living organisms [1]. He introduced the term Staphylococcus for the genus in 1882, and Rosenbach (1884) detached the genus into *S. aureus* and *S. albus* [2,3]. In 1939, Cowan distinguished *S. epidermidis* as a separate species based on coagulase testing [4]. Normally, *S. aureus* can also be found in healthy individuals [5]. It doesn’t cause any infection on healthy skin; however, upon entering the internal tissues or bloodstream, it may cause serious diseases [6]. *S. aureus* can cause minor skin infections such as impetigo, scalded skin, pimples, boils, abscesses, etc. It also causes life-threatening diseases such as meningitis, pneumonia, endocarditis, bacteremia, and sepsis [7,8,9,10].

According to studies, 20% of individuals are nasal carriers of *S. aureus*, and 30% are intermittent carriers [11]. As a result, *S. aureus* is one of the major reasons for the spread of hospital and community-acquired infections, which brings about serious consequences and life-threatening diseases. The discovery of antibiotics helped to treat the infectious diseases caused by *S. aureus*. The antibiotic penicillin was discovered in 1928 by Sir Alexander Fleming. The purified form of penicillin came in 1941 and saved the lives of many war victims during World War II. Within 2 years of introducing penicillin, *S. aureus* resistance had emerged [12]. The first penicillin-resistant *S. aureus* strain was identified in 1942 [13]. The semisynthetic antibiotic methicillin was designed in 1950s, and methicillin-resistant *S. aureus* (MRSA) was clinically detected in 1960s [14]. The first MRSA strains were found in the United Kingdom, and this epidemic was primarily constrained to Europe. Soon after, MRSA was identified in the United States, Japan, and Australia.

MRSA strains have the ability to produce a penicillin-binding protein (PBP) related with a diminished affinity for most semisynthetic penicillins. The PBP can be encoded by an obtained gene, mecA [15,16,17,18]. The methicillin-resistant genetic component (mecA) is carried on a mobile genetic element (MGE) selected staphylococcal cassette chromosome mec (SCCmec) [19,20]. The emergence of methicillin-resistant strains of staphylococci is because of the acquirement and inclusion of the mobile genetic elements into the chromosomes of the vulnerable strains.

### 1.2. Seriousness of MRSA

*S. aureus*, commonly known as “Staph” bacteria, is generally found on the skin or nose of about one-third of the inhabitants. These bacteria usually cause minor skin infections in healthy individuals. However, when *S. aureus* bacteria become resistant to any antibiotic, it can cause serious opportunistic infections or diseases. According to the survey taken by the US Centers for Disease Control and Prevention (CDC), around 5% of the populace carries the MRSA strain.

MRSA was first found in 1961. It is resistant to many antibiotics such as methicillin, penicillin, oxacillin, cloxacillin, cefazolin, cefoxitin, and other common antibiotics [21]. MRSA can spread via close contact with infected people. It can transmit from an object which contains MRSA to a human, or from a human carrier to another human. In general, any infections from hospitals or other healthcare centers are known as nosocomial infections. MRSA infection that occurs in hospital-associated/healthcare settings is termed as HA-MRSA. MRSA infection that occurs in community-associated settings is known as CA-MRSA. It can easily transmit between healthy people via skin-to-skin contact. CA-MRSA can easily affect elderly or debilitated people living in crowded conditions.

MRSA infections can resist the effects of numerous common antibiotics; hence, it is more intricate to treat. MRSA infections can influence the bloodstream, heart, bones, lungs, and joints of MRSA casualties. The world health organization (WHO) has listed MRSA as a “priority pathogen” due to its successful clones and ability to spread life-threatening diseases. Figure 1 represents the targets of antibiotics and the bacterial resistance mechanism against antibiotics.

In the 1980s and 1990s, the epidemic of MRSA spread across developed countries due to the rise of new clones. A study taken by Hetem and coworkers recommended that HA-MRSA clones tend to transfer at higher rates between hospitalized patients [22]. In 2008, over 380,000 HA-MRSA infections were acquired in European Union (EU) member states, Iceland, and Norway. The Brazilian clone (ST239-lll) is the major clone related to HA-MRSA infections and has been reported in various parts of Africa [23].

CC398 is the most widely recognized livestock-associated MRSA (LA-MRSA) strain worldwide. The origin of LA-MRSA seems to be related to pigs. CC398 shows little host specificity than other *S. aureus* isolates and moves simply between the hosts. CC398 is mainly associated with the food animal species such as cats, dogs, cows, poultry, and rabbits. It can also cause skeletal infections in broiler chickens. People in direct contact with livestock infected by MRSA, such as slaughterhouse workers, veterinarians and farmers, are at high risk of colonization with LA-MRSA.

Every year, 35,000 deaths occur in the United States because of antibiotic-resistant infections. The Public Health Agency of Canada (PHAC) has regarded MRSA as one of the priority antimicrobial-resistant pathogens. It can persistently evolve with a constant emergence of new genetically variant strains. Studies taken by the WHO reveal that people with MRSA infections are 64% more likely to die than people with other infections.

### 1.3. Current Threat

The spread of MRSA has been detected in every province of the planet. MRSA is currently endemic in most hospitals around the world. According to CDC assessments, MRSA has been reported as a serious threat after considering diverse factors, such as trouble on healthcare infrastructure and the community, prevalence, and escalating trends of resistance, treatability, mortality, preventability, and transmissibility. On the economic side, the annual healthcare costs for treating MRSA infections are estimated at $3 billion per year. CA-MRSA has been the principal MRSA infection in the past 10–20 years due to its successful clones [24].

The Asian Network for Surveillance of Resistant Pathogens (ANSORP) study shows MRSA infection rates of 38.1% for the Philippines, 57% for Thailand, and 74.1% for Vietnam in 2004–2006. Another report reveals that 18,000 patients lost their lives in the US during 2005, and the proportion of the mortality rate is higher than HIV/AIDS-associated fatalities.

The data collected from the Risk Stratification-based Surveillance (RSS) program in 2011 unveiled the truth that the MRSA ratio among clinical *S. aureus* isolates ranged from 28% in Indonesia and 59% in the Philippines. Studies taken by two multicenters in India showed MRSA infection proportions of 41% and 45%, respectively, in 2008–2009. A 2020 article revealed that MRSA commonly affects the skin and soft tissues. Bacteremia owing to *S. aureus* has been associated with 15% to 60% mortality rates. MRSA can also affect children to adults with no age restriction. MRSA bacteremia may lead to sudden death when there is no rapid diagnosis and treatment. According to some surveys taken in the US, up to 30% of children have MRSA on their skin or noses. Most children and adults didn’t know that they were MRSA carriers; hence, the bacteria live on their skin or in their noses cause serious complications upon relevant opportunistic health issues.

## 2. Current Therapeutic Strategies and Existing Drugs

### 2.1. Natural Extracts for MRSA Intervention

Recent studies concerning medicinal plants unveiled the truth that herbals can be a prolific source of antibacterial activities [25]. The herbals *G. mangostana* and *Q. infectoria*, in combination, has considerable activity against MRSA. Bassett et al., reported that tea-tree oil (TTO) shows favorable results in treating skin inflammation [26].

The phytochemicals in medicinal plants have various actions against resistant bacteria [27]. The multiple actions are:➢The increased penetrability of the cell wall membrane.➢Prohibition of the efflux pump systems.➢Modification of the active site.➢Enzymatic ruin and alteration of bacterial enzymes.

According to a study taken by Nassan et al., the aqueous extract of clove and cinnamon shows significant antibacterial activity against *E. coli* and *S. aureus* [28]. Badei et al., suggested that the combination of essential oils (cardamom, cinnamon, and clove) and phenol has antibacterial activity against 13 bacterial strains, 7 molds, and 2 yeasts [29]. Nature gives the necessary substrates to potentially fight against the pathogens that cause serious defects to humans.

### 2.2. Natural Drugs

Several natural products show potent antibacterial activity against *S. aureus*. Few natural compounds show potential bactericidal effects against drug-resistant *S. aureus*. The application of phytomedicines to treat drug-resistant bacterial infections has considerably increased worldwide in recent years.

Curcumin, Garlic, Ginger, Thai longan honey, Juncus and Luzula species, Greek oregano, Baru plant, and Lichen are natural products that show great activity against drug-resistant *S. aureus*. Apple cider vinegar has acidic as well as antimicrobial properties. Acetic acid is the main substance in vinegar. It can kill harmful bacteria or prevent them from multiplying. It is frequently used to treat eczema, dry skin, and skin infections.

Manuka honey has been suggested to be effective against MRSA because of its bactericidal activity. Recent studies unveiled that Manuka honey can inhibit more than twenty MRSA isolates at 4.4% (*v*/*v*) [30]. Another study shows that the growth of five MRSA strains was restrained by Manuka honey at 12.5% *v*/*v* dilution [31].

Malaysian tualang honey also has therapeutic potential, including wound recuperating properties and antimicrobial activity. Tualang honey can inhibit the growth of MRSA, *P. aeruginosa*, *S. aureus*, and *E. coli* with minimum inhibitory concentrations (MICs) comparable with those of Manuka honey [32].

Maureen U. Okwu et al., reported the anti-MRSA activities of some potential medicinal plants [33]. In this study, ethanol and methanol extracts of the medicinal plants were tested against the MRSA strains owing to the higher antimicrobial activity of alcoholic extracts than aqueous extracts (Table 1).

The phytochemical constituents present in the plants have shown good antibacterial activity against MRSA. For example, the flavonoids present in the plants form a complex with the bacterial cell wall, soluble protein, and extracellular substrates, and tannins inactivate envelop proteins and enzymes to lyse the bacteria.

### 2.3. Existing Synthetic Chemical Entities

Penicillin was initially used for the treatment of *S. aureus* infections. Primarily, methicillin is used to treat bacterial infections caused by *S. aureus*. Methicillin should be taken as an intravenous or intramuscular injection because it will be inactivated by gastric acid in the stomach when taken orally.

Vancomycin has been used for over 50 years to treat MRSA bacteremia. Vancomycin and its derivatives hamper the peptidoglycan (PG) biosynthesis of the Gram-positive bacteria by forming a strong hydrogen bond interaction with the D-Ala-A-Ala moiety of PG biosynthetic precursors [53].

Daptomycin is a cyclic lipopeptide, permitted in 2003 to be utilized for soft-tissue infections. It has a distinctive mode of action to kill bacteria which is concentration-dependent and works by binding to Gram-positive bacterial membranes. In vitro studies have proved that daptomycin has bacterial activity equivalent to or greater than vancomycin and linezolid [54].

Mupirocin is used to cure bacterial skin infections. It interferes with the isoleucyl t-RNA synthase in Gram-positive bacteria, inhibiting bacterial protein synthesis [55]. The recently launched drug linezolid helps to treat MRSA infections and other drug-resistant bacterial infections. Linezolid has a wide range of activity against MRSA, methicillin-resistant *S. epidermidis* (MRSE), vancomycin-resistant *S. aureus* (VRSA), and other Gram-positive pathogens; for example, penicillin-resistant streptococci and *Bacillus fragilis* [56]. Tigecycline, oritavancin, dalbavancin, iclaprim, cethromycin, and delafloxacin are semisynthetic drugs and show better activity against MRSA by inhibiting the peptidoglycan biosynthesis or protein synthesis.

### 2.4. Combining Synthetic and Natural Drugs

Synergistic or combination therapy is a new approach against resistant microorganisms such as MRSA, VRSA, etc. The use of plant extracts in combination with conventional antibiotics shows promising results in treating MRSA infections. The microdilution method or checkerboard method helps to determine the antimicrobial interaction between natural drugs and synthetic drugs. The alkaloids squalamine and lysergol, polyphenols, vitamins, and flavonoids have good synergistic effects against Gram-negative bacteria. Chiu-Fai Kuok et al., discussed the synergistic effect of natural products such as *M. officinalis*, *M. charantia*, *V. officinalis*, and *D. genkwa* with potential antibiotics [57]. This study unveiled that the tiliroside’s mechanism can inhibit MRSA’s penicillin-binding protein 2a (PBP2a) and penicillin-binding protein 4 (PBP4).

According to the study taken by Jency Blesson et al., the synergistic mixture of gentamicin and *C. esculenta* aqueous extract showed the maximum antibacterial activity against MRSA [58]. This study reveals that the bioactive compound present in the plant extract binds to the cell wall of the MRSA and increases the permeability of the cell wall. It enhances the flow of antibiotics into MRSA. The steroidal alkaloid tomatidine, produced by solanaceous plants, has potent antibacterial activity against *S. aureus* alone or with aminoglycosides [59]. Piperine is a bioactive compound present in pepper that shows good antibacterial activity against MRSA infections with the combination of gentamicin [60]. Synergism or a combination of drugs is a new concept in developing drug molecules and treating drug-resistant bacteria. The combination of drugs shows better results than the activity of individual medications. Hence, unique and effective drugs against resistant bacteria can be discovered via this method.

### 2.5. Multi-Drug Strategies

Combining two or more drugs shows more significant effects, and combination therapy has gained prevalent recognition, particularly in infectious diseases. As per the Safety and Innovation Act approved by the United States Congress in 2012, numerous antibiotics, for example, dalbavancin, tedizolid, and oritavancin were endorsed by the Food and Drug Administration (FDA). This act gives new impetus to the researchers via fast-tracked antibiotic endorsements and an additional 5 years of patent protection [61]. The Spanish Society of Clinical Microbiology and Infectious Diseases has already recommended combination therapy in certain conditions [62]. Researchers have also been examining the function of vancomycin or daptomycin in combination with β-lactam antibiotics. The combination therapy of antibiotics is widely used in clinical practice and it is being investigated as a novel approach in the fight against MRSA [63].

Linezolid, tigecycline, and daptomycin are the three novel antibacterial drugs for treating MRSA infections. Linezolid is a complete synthetic oxazolidinone that resists the initiation of protein synthesis in MRSA [64]. A study related to carbapenem with linezolid shows an excellent synergistic effect against MRSA infection. The combination of daptomycin and fosfomycin exhibits the highest synergistic effect against MRSA strains.

The daptomycin and oxacillin combination also shows a synergistic effect against MRSA strains [65]. A survey taken by Debbia et al., shows that daptomycin and gentamicin had a potential synergistic effect against *S. aureus* in a time-killing analyses [66]. Rifampicin and daptomycin combination has increased the MRSA biofilm clearance and showed a successful clinical rate in animal models [67]. Identifying the antimicrobial peptides (AMPs) to pair with conventional antibiotics shows promising results in treating *S. aureus* infections. The combination of Polysporin and Neosporin is used to treat skin infections caused by *S. aureus* [68]. Stan Deresinski discussed the treatment of serious MRSA infections using vancomycin in combination with other antibiotics [69]. In this study, the combination of vancomycin with a second antibiotic, rifampin or gentamicin, was suggested to treat MRSA infections.

Dilworth et al., illustrated a higher probability of destruction of MRSA from the bloodstream in patients treated with a blend of vancomycin and β-lactam antibiotic (*n* = 50) compared to those who obtained vancomycin alone (*n* = 30). Forty-eight patients (96%) in the combination therapy group and twenty-four patients (80%) in the vancomycin monotherapy group achieved abolition. In the subgroup of patients with endocarditis (*n* = 11 in each group), all 11 (100%) in the combination group and 9 (81.8%) in the vancomycin monotherapy group achieved microbiological abolition [70].

Antibiotic-loaded nanoparticles (NPs) have also been investigated by many researchers against resistant bacteria (Table 2). NPs have attractive antibacterial activity towards the resistant bacteria; hence more study is essential regarding the combinational methods, targeted and controlled release aspects of attaining resistance-free drugs [71].

### 2.6. Bacteriophage Therapy

Bacteriophage therapy, or phage therapy, is a cost-effective treatment method that uses viruses to treat bacterial infections. Phages can affect bacteria but are harmless to the populace, animals, and plants. Phage therapy has been proved as an effective treatment method for staphylococcal lung infections, *P. aeruginosa* infections, eye infections, and surgical wound infections. Phages are widely used in Russia and Georgia to treat bacterial infections that don’t respond to conventional antibiotics. Natalia et al., investigated the lytic activity of kayviruses against multidrug-resistant *S. aureus* [87]. They have studied morphology, genetics, biological properties, host range, latent time, phage burst size, adsorption rate, and lysis profiles of three phages. The morphological studies of the phages (vB_SauM-A, vB_SauM-C, and vB_SauM-D) showed a myovirion morphology, and the phage vB_SauM-A showed rapid adsorption, short latent period, and large burst size. Genomic studies revealed that the phages possess large genomes with low G+C content and similarity, like phage K.

### 2.7. Bacteriophage-Antibiotic Therapy

Phages are effective against treatable and resistant bacteria. They can prevent the bacteria from sharing antibiotic-resistant genes. Phages can be used alone or with antibiotics to treat multi-drug resistant bacterial infections. During the treatment, phages can multiply themselves and destroy the bacteria’s cell wall and cell membrane to kill the bacteria. Bartłomiej et al., discussed the bacteriophage-antibiotic therapy to treat *Acinetobacter baumannii* biofilm in a human urine model [88]. They used a cocktail composed of the Multi-Drug Resistant (MDR) *Acinetobacter baumannii* infecting bacteriophages in combination with antibiotics to demolish the bacterial biofilm in human urine. This study revealed that bacteriophage-antibiotic therapy could reduce biofilm biomass in a human urine model. The antibiotics used in the treatment showed an excellent synergistic effect with phage cocktails.

### 2.8. Vaccines for MRSA Treatment

The high antibiotic resistance profile of MRSA indicates the need for new interventions such as vaccines and new antibiotics. Translational science studies helps to assess vaccine candidates’ efficacy using animal models and in vitro and ex vivo models. In addition to bacteriophages, vaccines, monoclonal antibodies, and centyrins are being developed. Some of these strategies have been tested in humans with encouraging results. Clegg et al., reviewed the importance of identifying novel vaccine formulations that elicit potent humoral and cellular immune responses [89]. They have defined that the entering of vaccines into clinical trials offers the best success in treating MRSA infections, and a better understanding of the synergism of immunotherapies, antibiotics, and vaccines can help to design future clinical trials.

## 3. Current Treatment Approaches and Advances

### 3.1. Chemotherapeutics for MRSA

Chemotherapy is a drug treatment generally used to kill fast-growing cells in the human body using powerful chemicals. Chemotherapy drugs can be used alone or combined with other medications to treat various diseases or infections. Patients with malignancy, especially hematological malignancy, are at high risk of MRSA infection. Vancomycin is the primary antibiotic for the treatment of MRSA infections.

Teicoplanin, telavancin, ceftaroline, daptomycin, oxazolidinones, and vancomycin are the prominent antibiotics widely used to treat MRSA infections. To treat MRSA infections, a combination of chemotherapy with vancomycin or arbekacin plus a β-lactam antibiotic is recommended, and a granulocyte colony-stimulating factor is clinically helpful when the granulocytopenia was induced by chemotherapy. Hiramatsu et al., found that nybomycin can strongly obstruct the function of the mutated DNA gyrase of quinolone-susceptible *S. aureus* [90]. Recent studies unveiled the truth that pleuromutilins can be used to treat impetigo caused by methicillin-susceptible *S. aureus* (MSSA) and *Streptococcus pyogenes*.

Jacqueline C et al., reported the most active bactericidal drug of ceftaroline fosamil, in the bunny model of MRSA endocarditis [91]. For each MRSA strain, bunnies were randomized to no treatment (controls), a ceftaroline fosamil dose comparable to 10 mg/kg/12 h in human beings (600 mg twice daily), daptomycin at a dose equivalent to 6 mg/kg/24 h in human beings, or a tigecycline dose equal to 100 mg/24 h in humans plus 50 mg/12 h. Ceftaroline and daptomycin showed high bactericidal efficiency based on MRSA growth diminution rates (>5 log_10_ CFU/g). In contrast, tigecycline didn’t show bactericidal efficiency, and MRSA growth diminution rates were <2 log_10_ CFC/g compared to controls. However, ceftaroline’s MRSA sterilization rate was 100% compared to the 57% attained by daptomycin, with resistant mutants only seen in the daptomycin treatment group.

Linezolid has also been reported as an effective drug against MRSA bacteremia [92]. In a prospective open randomized trial, clinical triumph at the cure analysis was achieved in 19 of 24 (79.2%) linezolid recipients and 16 of 21 (76.2%) vancomycin recipients. Tsai et al., investigated the efficacy of teicoplanin against MRSA bacteremia [93]. A total of 146 adult patients with MRSA bacteremia were treated with teicoplanin. In that case, a considerable number of patients in the high-dose regimen group (6 mg/kg/12 h) showed favorable outcomes compared to patients treated with a standard-dose (6 mg/kg/24 h) (84.1% vs. 41.2%; *p* < 0.01). This study proposed the high-dose teicoplanin treatment for MRSA bacteremia rather than the standard dose of teicoplanin treatment.

Stryjewski et al., discussed the treatment of complicated skin and soft-tissue infections caused by Gram-positive bacteria using telavancin versus standard therapy [94]. In this case, 167 patients were selected and given at least one dose of study medication. Of the patients with *S. aureus* infection at baseline (*n* = 102), 80% of the patients (telavancin group) were cured, and 77% of the patients in the standard therapy group were cured. Cure rates of 82% and 69% were shown in the telavancin and traditional therapy groups, respectively, for patients with MRSA infection at baseline (*n* = 48).

### 3.2. Nanomedicinal Platforms

The WHO states that antibiotic-resistant bacteria are one of the most extraordinary obstacles to global health and progress. Nanomaterial approaches to fight against MRSA are the emerging paradigm that shows the challenges faced by humans due to infections caused by resistant bacteria [95]. Specific NPs can fight with resistant bacteria via several mechanisms.

In April 2011, IBM and Institute of Bioengineering and Nanotechnology (IBN) researchers discovered a new type of polymer that can detect and destroy resistant bacteria and infectious diseases such as MRSA. These biodegradable nanostructures can be infused directly into the human body or applied topically to the skin.

Mamun et al., reported the functions and properties of nanoantibiotics to combat antibiotic resistance [96]. Due to having a high surface-to-volume ratio and specific surface region, NPs possess an extremely high contact area than molecular materials of a similar mass. Organic NPs (ONPs) synthesized from organic compounds such as proteins, lipids, carbohydrates, nucleic acids, etc., can directly interact with the bioactive components of bacteria. The metallic NPs and ONPs can concurrently exhibit three distinct antimicrobial mechanisms when functionalized with antibiotics.

The oxidative mechanism includes: stress induction through reactive oxygen species (ROS) and free-radical generation, electron transport chain restraint, plasmid damage, disruption of the cell membrane, DNA damage, interruption of enzyme activity, metal ion release including the crumbling of metal ions from nanoconjugates into dissolve form, interruption of the electron transport chain, and non-oxidative attributes such as surface energy, size, shape, surface roughness, cytoplasm release, atomically thin structures, zeta potential (surface charge), stability, increased specific surface region/volume ratio, high surface reactivity, conductivity, the bioavailability of NPs, drug release systems at target sites, physical interactions between particles, and cell wall/membranes and particle-antibiotic interface, etc. These features of NPs permit appropriate contact with bacterial cells and improved penetration capacities crossing the cell membrane while meddling with cellular elements and metabolic machinery. Figure 2 represents how nanoscale particles function as carriers of drug molecules across bacterial membrane barriers.

Daniel Hasan et al., reported the formulation of quatsomes for enhanced delivery of vancomycin against MRSA [97]. The responsive quatsomes were prepared from pH-responsive quaternary bicephalic surfactants (StBAclm) and cholesterol (Chol). Drug-loaded quatsomes were developed by adding the bicephalic surfactant, vancomycin, and Chol at different concentrations. This was followed by adding 40 mL of distilled water, and the mixture was further sonicated for 10, 15, 20, and 30 min using a probe-sonicator in an ice bath. The dispersions were further stirred at 500 rpm using a magnetic stirrer for 24 h at room temperature. After 24 h of equilibration, the potential aggregates were detached from the blank (StBAclm-Qt) and the drug-loaded vancomycin-StBAclm-Qt quatsomes using a filter/syringe. These all showed an enhanced antimicrobial effect of the drug-loaded quatsomes at an eight-fold lower concentration when compared to bare vancomycin. The drug-loaded quatsomes also demonstrated the capability to abolish MRSA biofilm.

Giersing et al., discussed nano-patterning technologies and surface alteration to deter MRSA biofilm formation [98]. The drug-loaded NPs could overcome the limitations of conventional antibiotic treatments. Metal containing NPs have the potential to inhibit the growth of MRSA infections. When the positively charged ions of the NPs combine with the negatively charged parts of the bacterial cell membrane, the bacterial cell wall or membrane disruption occurs. This creates pores in the membrane and leads to cell death by dissipating the H^+^ gradient across the membrane [99]. ROS formation results in DNA damage and cell fatality when the NP’s internalization occurs in the bacterial cytosol [100]. The development of nanotechnology gives a promising solution for treating resistant bacterial infections.

### 3.3. Antimicrobial Activity of Surfactant Molecules

Generally, the antimicrobial activity of surfactants depends on the alkyl chain length, and surfactants show increased antibacterial activity while having decreased chain length. Cationic surfactants such as quaternary ammonium compounds show good antibacterial activity because they have high cationic charge and hydrophobicity. The positive control of cationic surfactants leads to a more efficient interaction with the negatively charged bacterial membrane. Robert et al., discussed the effectiveness of humimycins, N-acylated linear heptapeptides, against MRSA [101]. They have altered the structure of the lipopeptides, including hydrophobicity and enantiomeric amino acid substitutions, to explore their potential against MRSA. This study revealed that the dihydroxy analogue, synthesized with myristic anhydride, shows four times better activity against MRSA in minimum inhibitory concentration assays.

### 3.4. Radiation Therapy

Radiation therapy is generally used for cancer treatment that uses intense beam energy or photons, or other types of energy, to kill cancer cells. During brachytherapy, radiation will be placed inside the human body. Pulse laser therapy effectively kills MRSA and shortens the therapy time. It doesn’t give any heat or pain sensation; hence, it is ideal for clinical applications. Diabetic patients with open wounds are highly susceptible to MRSA when the infection takes hold and creates a biofilm or a slimy buildup of bacteria that become more difficult to treat.

Boston University College of Engineering researchers invented a new technique using radiation therapy that can kill 99.9% of MRSA [102]. The golden pigment is the universal signature of *S. aureus*. MRSA is exposed to blue light that causes a traumatic photobleaching effect and destroys the pigment molecule responsible for the cell’s golden color. Vulnerable spots will appear on the cell membrane when the golden pigment molecules are decomposed under blue light. The blue light can destabilize MRSA cells and enough to kill about 90% of the bacterial culture. The powerful oxidizing agent H_2_O_2_ can kill the remaining, and critical, 10%. The research in mice using blue light and H_2_O_2_ to treat MRSA infections revealed that this method could speed up the healing of skin wounds infected with MRSA.

## 4. In Vitro and In Vivo Studies

MRSA isolates extracellular matrix production is the critical virulence factor and survival mechanism for this superbug [103]. An in vitro study refers to work performed in a controlled environment such as a test tube or petri dish. In vitro studies can be achieved only outside of a living organism. In vivo research can be done with or within an entire living organism. Studies in animal models or human clinical trials are in vivo studies.

There is no coercion to submit animal protocols to Institutional Animal Care and Use Committee (IACUCs) for doing in vitro studies [104]. This method helps to reduce the requirement for experienced laboratory personnel in animal handling. The loss of glycosylation of the antibody might make the antibody product unsuitable for in vivo experiments.

Mohammad et al., examined the potential of thiazole compounds against MRSA [105]. They have reported that phenylthiazole compounds can exhibit effective antimicrobial activity in in vitro studies against various arrays of clinically meaningful MRSA strains. Derivatives of the compound were synthesized, and the structure-activity relationship has been elucidated. Figure 3 represents the Chemical structures of thiazole compounds.

These derivatives unveiled that the aminoguanidine moiety at thiazole-C5 is essential for the antibacterial activity. A nonpolar, hydrophobic group is preferential at thiazole-C2. Analogs to the lipophilic alkyl tail of phenylthiazole compounds were subsequently developed to improve the antimicrobial activity of the thiazole compounds, toxicity profile, and their physicochemical properties. These phenylthiazole compounds have numerous admirable characteristics in vitro, including quick bactericidal activity against MRSA, a low frequency of bacterial resistance developing, and they have exposed the possibility of being used in a blend with other antibiotics, such as vancomycin, against MRSA. The compounds (**1**–**5**) were tested (at concentrations of 5 µg/mL, 10 µg/mL, 20 µg/mL, and 40 µg/mL) against a human keratinocyte (HaCaT) cell line to determine the potential toxic effect to mammalian skin cells using in vitro analysis.

The cells were cultured in Dulbecco’s modified Eagle medium (DMEM) supplemented with 10% fetal bovine serum (FBS) at 37 °C with CO_2_ (5%). Control cells obtained Dimethyl sulfoxide (DMSO) alone at a concentration equivalent to that in drug-treated cell samples. The cells were incubated with the compounds (in triplicate) in a 96-well plate at 37 °C with CO_2_ (5%) for two hours before the addition of the assay reagent MTS 3-(4,5-dimethylthiazole-2-yl)-5-(3-carboxymethoxyphenyl)-2-(4-sulfophenyl)-2*H*-tetrazolium). Absorbance readings (at OD_490_) were calculated using a kinetic microplate reader. After treatment with each compound, the extent number of viable cells was expressed as a percentage of the viability of DMSO-treated control cells. The toxicity data were examined through a one-way Analysis of variance (ANOVA), with post hoc Dunnett’s multiple comparisons tests (*p* < 0.05), utilizing GraphPad Prism 6.0. In vitro studies revealed that the thiazole compounds are nontoxic to human keratinocytes and negatively affect mammalian tissues. Furthermore, the antimicrobial activities of the thiazole compounds were confirmed by performing an in vivo analysis.

The in vivo process evades the necessity to teach the antibody producer tissue-culture methods. When performing the in vivo method (i.e., mouse ascites method), a very high mAb concentration will be produced for this reason, there is no need for additional concentration procedures, which can denature the antibody and reduce efficiency. In vivo strategies can create significant pain or trouble in mice. The mAb produced via in vivo methods can contain diverse contaminations that might need purification.

The in vivo studies were performed with eight groups (*n* = 5) of eight-week-old female Balb/c mice. The mice were disinfected with ethanol (70%) and shaved on the back center (roughly a one-inch by one-inch square region around the injection site) one day before infection. The bacterial inoculum preparation, an aliquot of an overnight culture of MRSA USA300, was transferred to a clean tryptic soy broth (TSB) and shaken at 37 °C until an OD_600_ value of ~1.0 was accomplished.

The cells were centrifuged, washed once with phosphate buffer solution (PBS), re-centrifuged, and then resuspended in PBS. The mice were given an intradermal injection (20 µL) containing ~2.76 × 10^8^ CFU/mL MRSA USA300. An open injury was created at the injection site, 48 h post-infection. Topical treatment was initiated consequently with each group of mice getting thiazole compounds (2%, using petroleum jelly as the vehicle), mupirocin (2%, using petroleum jelly as the vehicle), thiazole compound-**1** (2%, using Lipoderm as an alternative vehicle) and a control group obtaining the control vehicle (20 mg, petroleum jelly) alone. Each group of mice receiving a particular treatment procedure was housed individually in a ventilated cage with proper bedding, food, and water. The mice were tested twice a day during infection and treatment to ensure there were no unfavorable reactions. The mice were humanely euthanized by CO_2_ asphyxiation 24 h after administering the last dose. The region around the skin injury was delicately swabbed with ethanol (70%) and expelled. Then, the tissue was homogenized in 1 mL of TSB. The homogenized tissue was consecutively diluted in PBS before plating onto mannitol salt agar plates. The plates were incubated for 20–22 h at 37 °C before feasible CFCs were counted, and MRSA diminution in the skin injury post-treatment was determined for each group. Using a one-way ANOVA, data were analyzed with post hoc Holm-Sidak’s multiple comparison tests (*p* < 0.05), utilizing GraphPad Prism 6.0.

Thiazole-1 showed effective activity against all drug-resistant staphylococcal strains with the MIC value of 1.3 µg/mL. The biphenyl and butyne analogs (2 and 3, respectively) have MIC values ranging from 2.8 to 5.6 µg/µL. All five thiazole compounds have potent antimicrobial activity against MRSA strains that exhibit resistance to antibiotics, including β-lactams, fluoroquinolones (USA800), tetracycline (USA300), and erythromycin (USA300 and USA1000).

These compounds also exhibited intense antimicrobial activity against *S. aureus* NRS107 (MIC values range from 1.3 to 13.3 µg/mL), a strain showing high resistance to mupirocin (MIC of 1024.0 µg/mL). Moreover, compounds **1** and **2** (MIC of 1.3 and 2.8 µg/mL, respectively) are more active than mupirocin (MIC of 4.0 µg/mL) against three additional MRSA strains (USA800, USA1000, and USA1100). When tested against four MRSA strains, Clindamycin was determined to have a 0.1 µg/mL MIC. The results showed that the thiazole compounds have intense activity against CA-MRSA strains and MRSA isolates liable for infected wounds in patients (Table 3).

In this study, the antimicrobial activity of phenylthiazole compounds has been illustrated using in vitro and in vivo analysis against clinically pertinent MRSA strains liable for skin and wound infections. The thiazole compounds were reported as nontoxic to human keratinocytes and exhibited potent antimicrobial activity against MRSA.

## 5. Clinical Analysis

Clinical analysis is performed via blood, plasma, serum, urine, fluids, tissues, and occasionally on other matrices. Generally, analytes are measured in serum due to their easy access, and it is used to reflect target concentrations in tissue or receptors. Intensive care patients have a higher risk compared with medical patients. They have a higher rate of developing MRSA infection within the first 4 days of admission.

Patients with surgical wounds, ulcers, and intravenous catheterization also have risk factors, with hazard ratios of 2.9, 3.0, and 4.7 [106]. Protection of patients with ulcers, intravenous catheters, and surgical wounds from MRSA outbreaks helps to reduce the spread and seriousness of MRSA infections. Using antibiotics, herbals, radiation therapy, or other treatment methods may help patients to recover from MRSA infections. Proper knowledge about MRSA, earlier treatment, and reducing the risk factors can also help to decrease the mortality rate of MRSA infections.

Jang et al., assessed the patients who had been suffering from *S. aureus* bacteremia [107]. Among the 377 patients with *S. aureus* bacteremia, 41 cases (11%) were persistent even with proper antibiotics. Of these, 35 of the 41 persistent bacteremia patients had persistence due to MRSA. Salvage treatment comprised of either adding an aminoglycoside or rifampicin to vancomycin or a vancomycin substitution with linezolid alone or blending with a carbapenem. This study revealed that linezolid substitution was considerably more efficient, with MRSA clearance in 75% of cases compared with 17% with vancomycin-based treatment continuation.

Lai et al., estimated the use of high-dose daptomycin in patients with severe Gram-positive infections, with 63 out of 67 (94%) patients getting high-dose daptomycin as a salvage therapy. MRSA was segregated in 56.7% of patients, and 80.6% of patients were bacteremic. The clinical and microbiological triumph rates were 77.6% and 82.1%, respectively [108].

Caelli et al., compared the use of a combination of 4% TTO nasal ointment and 5% TTO body wash (intervention) with a standard 2% mupirocin nasal ointment and triclosan body wash (routine) for the annihilation of MRSA [109]. A total of 30 in-patients were selected and erratically assigned to be treated with TTO or standard routine care for 3 days. The patients also obtained intravenous vancomycin, and all the patients were evaluated for MRSA carriage after 48 and 96 h of the discontinuance of topical treatment. More patients in the intervention than in the control group were cleared of infection (5/8 versus 2/10). Two patients in the intervention group were treated for 34 days and removed the infection, while the other remained chronically colonized.

## 6. Computational Analysis

Computer-aided design is a preliminary stand for screening novel inhibitors, and the statistics help to find new applications in drug development. Computational analysis helps to understand mainly by studying mathematical models implemented on computers. Computation is any type of calculation that includes arithmetical and non-arithmetical processes and pursues a well-defined model. Computational analysis helps researchers to carry out thousands of simulated experiments.

Finite Element Method (FEM) is a computational analysis that is widely used in orthopedic biomechanics [110]. Various computational methods are available to identify the best compound for the treatment of MRSA infections, including docking studies, ADMET analysis, and the characterization of the respective mechanism. The target protein molecules can be designed via homology modeling or collected from the Protein Data Bank (PDB) database and the potential drug molecules suggested from the PubChem database [111]. Computational models are used to track contagious diseases in populaces that help to identify the most efficient interventions and adjust interventions to diminish the spread of diseases. Scientists use computational modeling to design the safest drugs for patients with minor side effects.

Skariyachan et al., has reported the computer-aided screening and assessment of herbal therapeutics against MRSA infections [112]. In this study, absorption, distribution, metabolism, excretion, toxicity (ADMET) and docking studies were used to screen the drug-likeness and effectiveness of diverse herbal compounds. The structures of the proteins PBP2A and Panton-Valentine Leukocidin (PVL) were used as the drug targets. The 3D structures of all plant-based ligands were collected from the drug database, and the structures were reclaimed from Kyoto Encyclopedia of Genes and Genomes (KEGG), PubChem, and Chemspider.

Computational biology tools can predict the molecules’ absorption, distribution, metabolism, excretion, and toxicity (ADMET). The 74 ligands identified for this study have been analyzed for their drug-likeliness, ADME properties, and toxicity investigation by Pre-ADMET. The ADME includes the extent and rate of absorption, distribution, metabolism, and excretion. Pre-ADMET uses Caco2-cell (heterogeneous human epithelial colorectal adenocarcinoma cell lines) and MDCK (Madlin-Darby Canine Kidney) cell models for oral drug absorption prediction, and skin penetrability and human gastrointestinal absorption models for oral and transdermal drug absorption prediction. Pre-ADMET helps to predict toxicity based on the mutagenicity of AMES parameters and rodent carcinogenicity assays of rats and mice. The crystal structures of PBP2A (1VQQ) and PVL toxin (1T5R) were reclaimed from the Protein Data Bank, and the harmonized files were used for the study. The chosen herbal compounds were docked with the target proteins by AutoDock. The program uses a Monte Carlo replicated annealing for configurational investigation using grid-based molecular affinity potentials and confers bioactive conformation by energy minimization.

The docking studies unveiled a truth that the bioactive constituent of Aloe vera, β-sitosterol (3S,8S,9S,10R,13R,14S,17R) 17 [(2R,5R)-5-ethyl-6-methylheptan-2-yl]-10, 13-dimethyl 2,3,4,7,8,9,11,12,14,15,16,17-dodecahydro-1*H*-cyclopenta [a] phenanthren-3-ol) exhibited excellent binding energies of −7.40 kcal/mol and −6.34 kcal/mol for PBP2A and PVL toxin, respectively. In the same way, Meliantriol (1S-1-[(2R,3R, 5R)-5-hydorxy-3-[(3S,5R,9R,10R,13S,14S,17S)-3-hydroxy 4,4,10,13,14-pentamethyl-2,3,5, 6,9,11,12,15,16,17-decahydro-1*H*-cyclopenta [a] phenanthren-17-yl] oxolan-2-yl]-2-methylpropane-1,2 diol), bioactive compound in Azadirachta indica (Neem) exhibited the binding energies of −6.02 kcal/mol for PBP2A and −8.94 for PVL toxin. The in-silico and in vitro studies revealed that the herbal extracts of Aloevera, Guava, Neem, Pomegranate, and Tea can be used as therapeutics against MRSA infections.

### Molecular Dynamics

Traditional docking methods have significant limitations related to the semi-flexible treatment of ligands (small molecules) and targets (proteins) [113]. Atomistic molecular dynamics (MDs) have emerged as an alternative for simulating macromolecular complexes. MDs helps to explore drug-target recognition and binding from mechanical and energetic perspectives and determine the binding and unbinding kinetic constants. Many computational works have been developed based on combined docking MD strategies. Solvent mapping or co-solvent MD simulation techniques have been widely used to identify binding sites and hot spots on protein surfaces at a dynamic level.

MD simulation helps to reveal unstable binding modes according to the structural perspective. This technique helps to negate physically unreliable docking solutions or helps to find new ones [114]. Post-processing via MD helps to refine the energy assessed by scoring functions (re-scoring). A wide range of theoretical methods are needed for re-scoring approaches, and the energies are computed as ensemble averages [115]. Re-scoring techniques range from partially empirical methods such as Linear Interaction Energy through an authentic free-energy process. The probable solution may lie between the Molecular Mechanics-Poisson-Boltzmann Surface Area and the Molecular Mechanics-Generalized Born Surface Area re-scoring approach, which helps to balance reasonable accuracy and computational costs [116].

Another method has been used to generate an ensemble of protein conformations by performing MD before docking. In this method, the targeting flexibility in docking will be taken into account indirectly and Virtual Screening (VS) through pre-generated discrete conformations. McCammon and co-workers implemented the “relaxed complex scheme”, which helps to perform extensive MD simulation of the protein in the apo form [117,118]. Then, snapshots were taken at regular time intervals, and each protein targeted in the subsequent docking exercise. The problem with this method is choosing the appropriate protein structures to be used to build the ensemble. A large number of protein conformations may worsen the VS performances. Cluster analysis techniques can help to reduce redundancy in the conformation set and ensure that only dissimilar structures are included. Enhanced sampling techniques can be used in this method to further increase the protein conformations’ variability [119].

Dynamic docking methods are different from the other methods due to the possibility of characterizing the protein-ligand binding process. Dynamic docking exploits the advances in sampling strategies. There are two conceptual layers of complexity in the static, dynamic method. The first layer is related to engendering the binding modes, and the second layer is about evaluating the reliability of the identified “poses.” This can be achieved by considering the binding energy. Dynamic docking methods are helpful in the drug discovery program, and feasible advances may happen in the future by improving hardware architecture and machine-learning techniques.

## 7. Future Directions for MRSA Treatment

MRSA infections are continuously growing burden for human society due to the poor prognosis and high costs associated with this infection [120]. Most of the antibiotics available in the market become less active against resistant bacteria, particularly against MRSA; hence, new strategies are needed to treat MRSA infections.

The future treatment methods for MRSA infection may have the following aspects:Nanocarriers with good surface area for controlled release of antibiotic with minimum inhibitory concentrations.The design and development of antibody-based biologic agent treatments for the management of serious infections of MRSA.Multi-drug strategies for treating drug-resistant bacterial infections such as synthetic drugs, synthetic and natural drugs, and natural medicines.Breakage of biofilms formed by MRSA using a suitable targeting carrier system with the biotic drugs.

## 8. Conclusions

Many lives have been saved from numerous contagious diseases by the discovery of antibiotics, but MRSA infections are formidable, versatile, and unpredictable. Vancomycin is mainly used to treat MRSA infections, either alone or in combination. However, intermediate resistance levels, vigilant therapeutic drug monitoring, and MIC creep are the path blockers for vancomycin against MRSA infections. Improving efficient anti-MRSA treatments and the execution of infection control methodologies are of expeditious need. Advanced research in this field may reduce the clinical impact of this pathogen. The combination of antibiotics with NPs can be a precedent shift to distribute the ideal concentration of a drug in the target place by bypassing resistance mechanisms of bacteria. The combination of antibiotics with NPs can accomplish all the requirements of an effective antibacterial agent and save millions of lives. The efficiency and biosafety of the designed and synthesized drugs can be studied using in vivo and ex vivo analyses. These studies may reduce the difficulties in human trials.

## Figures and Tables

**Figure 1 antibiotics-11-00606-f001:**
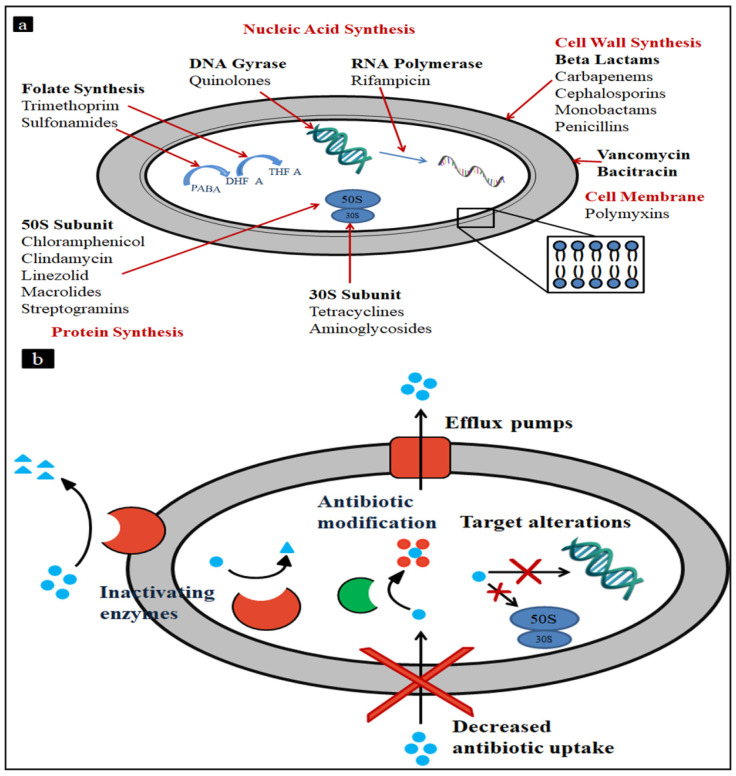
(**a**) Antibiotic targets and (**b**) bacterial resistance mechanism.

**Figure 2 antibiotics-11-00606-f002:**
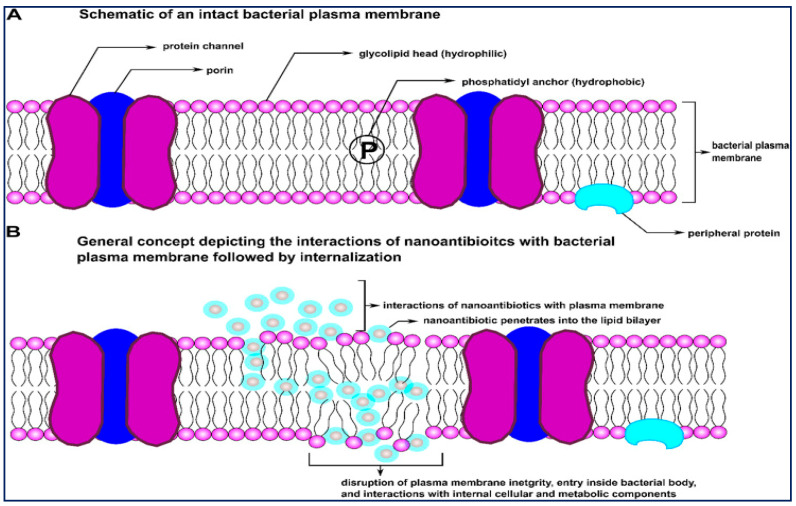
Schematic representation of (**A**) an intact bacterial cell membrane and (**B**) the effect of nanoantibiotics on the reliability of a bacterial cell membrane (Adapted from Ref. [96]).

**Figure 3 antibiotics-11-00606-f003:**
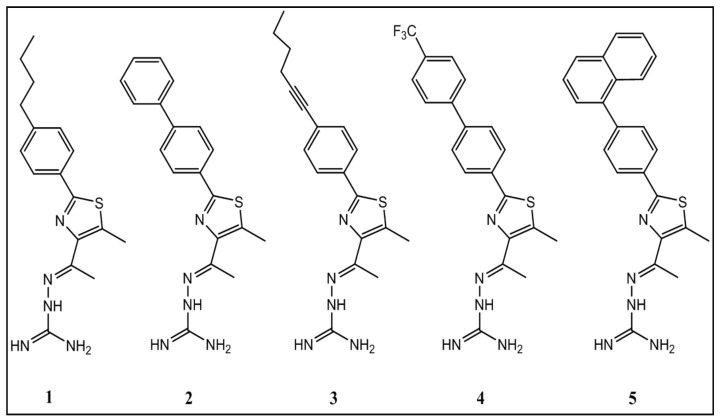
Chemical structures of thiazole compounds 1–5 (Adapted from Ref. [105]).

**Table 1 antibiotics-11-00606-t001:** Anti-MRSA activities of medicinal plants.

S. No	Botanical Name	Common Name	Plant Part Used	Extracting Solvent	MIC/MBC (mg/mL) MRSA	Ref.
1.	*Butea monosperma Lam.*	Flame-of-the-forest	Leaf	Ethanol	5.91/13.30	[34]
2.	*Acacia catechu (L. f.) Willd*	Cutch tree, black catechu	Wood	Ethanol	1.6–3.2/25	[35]
3.	*Callistemon rigidus R.Br.*	Stiff bottlebrush	Leaf	Methanol	0.00125–0.08	[36]
4.	*Acacia albida Del.*	Gawo	Stem bark	Methanol	3.0/4.0	[37]
5.	*Nymphaea lotus Linn.*	White lotus	Leaf	Ethanol	5.0–10.0/10.0–30.0	[38]
6.	*Impatiens balsamina*	Garden balsam	Leaf	Ethanol	6.3/25	[39]
7.	*Garcinia mangostana* L.	Mangosteen	Fruit shell	Ethanol	0.05–0.4/0.1–0.4	[40]
8.	*Peltophorum pterocarpum (DC.)*	Yellow flame tree	Bark	Ethanol	0.1–0.8/6.3	[41]
9.	*Psidium guajava* L.	Guava	Leaf	Ethanol	0.2–1.6/6.3	[42]
10.	*Punica granatum* L.	Pomegranate	Fruit shell	Ethanol	0.2–0.4/1.6–3.2	[43]
11.	*Uncaria gambir (Hunter) Roxb.*	Gambier, White cutch	Leaf, stem	Ethanol	0.4–0.8/3.2	[44]
12.	*Walsura robusta*	Bonlichu	Wood	Ethanol	1.6–3.2/25	[45]
13.	*Swietenia mahagoni*	Mahagoni	Seed	Ethanol	0.2–0.78/0.78–1.56	[46]
14.	*Senna alata*	Candle bush	Leaf	Ethanol	0.5/ND	[47]
15.	*Garcinia Morella Desr.*	Gamboge	Whole plant	Ethanol	0.016–0.064/0.064–0.26	[48]
16.	*Melianthus major* L.	Giant honey flower	Leaves	Ethanol	0.78/3.12	[49]
17.	*Melianthus comosus Vahl*	Honey flower	Leaves	Ethanol	0.39/1.56	[49]
18.	*Withania somnifera* L.	Ashwagandha	Roots & leaves	Ethanol	1.56/>6.25	[50]
19.	*Quercus infectoria Olivier*	Oak galls	Nutgalls	Ethanol	0.4–3.2/3.2–6.3	[51]
20.	*Thymus vulgaris* L.	Thyme	Leaves	Essential oil	0.057/ND	[52]

**Table 2 antibiotics-11-00606-t002:** The synergy of plant-based or microbial-based NPs and antibiotics for antibacterial activity.

Nano Particles	Source of Reducing Agent	Combination of Antibiotic	Targeted Bacteria	Ref.
Ag NP	Leaf extract of *Typha angustifolia*	Gentamicin, cefotaxime, meropenem	*E. coli*, *K. pneumonia*	[72]
Ag NP	*Dioscorea bulbifera* tuber extract	β-lactam (piperacillin) and macrolide (erythromycin)	*A. baumannii*	[73]
Ag NP	Silky hairs of aqueous corn extract	Kanamycin and rifampicin	*E. coli*, *S. aureus*	[74]
Ag NP	*Eichhornia crassipes*	Vancomycin, penicillin, streptomycin, and tetracycline	*S. aureus*, *E. coli*, *K. pneumonia*, *Enterococcus*	[75]
Au NP	*Citrullus lanatus* rind extract	Kanamycin, rifampicin	*Bacillus cereus*, *E. coli*, *Listeria monocytogenes*, *S. aureus*, *S. typhi*	[76]
Ag NP	Corn leaf waste of *Zea mays* extract	Kanamycin and rifampicin	*E. coli* ATCC 42890, *Bacillus cereus* ATCC 13061 19115, and *S. aureus* ATCC 49444	[77]
Ag NP	Flower broth of *Tagetes erecta*	Commercial antibiotics	Gram-positive (*S. aureus* and *B. cereus*), Gram-negative (*E. coli* and *P. aeruginosa*) bacteria	[78]
Ag NP	Gum kondagogu	Ciprofloxacin, streptomycin, and gentamycin	Gram-positive (*S. aureus* 25923, *S. aureus* 49834) and Gram-negative (*E. coli* 25922, *P. aeruginosa* 27853)	[79]
Ag NP	*Acinetobacter calcoaceticus*	Aminoglycosides, β-lactams, cephalosporins, tetracyclines	*A. baumannii*, *P. aeruginosa*, *S. aureus*, *S. typhi*	[80]
Ag NP	*E. coli*	Bacitracin, ampicillin, kanamycin	*Corynebacterium diphtheria*, *E. coli*, *K. pneumonia*	[81]
Ag NP	*Aspergillus flavus* and *Emericella nidulans*	Amikacin, kanamycin and streptomycin	*E. coli*, *S. aureus* and *P. aeruginosa*	[82]
Ag NP	*Aspergillus flavus*	Ciprofloxacin, vancomycin, gentamycin	Gram-positive and Gram-negative bacteria	[83]
Ag NP	Bacteria from petroleum soil	Doxycycline	*Klebsiella pneumonia*	[84]
Ag NP	*Streptomyces xinghaiensis* OF1 strain	Ampicillin, kanamycin, and tetracycline	*E. coli*, *P. aeruginosa*, *S. aureus* and *Klebsiella pneumonia*	[85]
Ag NP	*Trichoderma viride*	Ampicillin, kanamycin, and erythromycin	Gram-positive and Gram-negative bacteria	[86]

**Table 3 antibiotics-11-00606-t003:** Minimum inhibitory concentration (MIC in µg/mL) of thiazole compounds **1**–**5**, clindamycin and mupirocin (tested in triplicate) against MRSA and mupirocin-resistant *S. aureus* (NRS107) strain isolated from skin wounds [105].

Compound	*S. aureus* Strain Number					
	NRS107	USA300	USA400	USA800	USA1000	USA1100
**1**	1.3	1.3	1.3	1.3	1.3	1.3
**2**	2.8	2.8	2.8	2.8	2.8	2.8
**3**	2.8	5.6	5.6	5.6	2.8	5.6
**4**	13.3	13.3	13.3	13.3	13.3	13.3
**5**	6.4	6.4	12.8	6.4	12.8	6.4
Mupirocin	1024.0	1.0	1.0	4.0	4.0	4.0
Clindamycin	0.1	1.8	0.1	0.1	0.1	0.1

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
