# Peer review of "Recent Developments in Methicillin-Resistant *Staphylococcus aureus* (MRSA) Treatment: A Review"

_antibiotics, 2022, doi:10.3390/antibiotics11050606_

Round 1

Reviewer 1 Report

As the authors correctly identify the risks associated with MRSA, the work as a whole is interesting. You should definitely consider whether the layout of the work is logical enough and whether the text is fluent - some paragraphs do not flow smoothly one of the other. I am also missing a summary of bacteriophage therapy. Below are works that can help in the preparation of such a study:

https://www.mdpi.com/2076-2607/7/10/471

an interesting approach is also a combined bacteriophage-antibiotic therapy, little has been done here on the staphylococcal model, but there are other works available:

https://www.liebertpub.com/doi/abs/10.1089/mdr.2020.0083

Author Response

Author’s Reply to Editor and Reviewers Comments

Journal                       : Antibiotics

Manuscript Title        : Recent Developments in Methicillin-Resistant Staphylococcus aureus (MRSA) Treatment: A Review

Manuscript Ref. No. : 1607808

Authors                      : Palanichamy Nandhini1, Pradeep Kumar2, Suresh Mickymaray3, Abdulaziz Al othaim3, Jayaprakash Somasundaram4 and Mariappan Rajan1,*

COMMENTS FROM THE AUTHOR:

The authors thank the editor and reviewers for the valuable comments and suggestions. As far as possible, we have answered the comments and suggestions given by the reviewers.

The changes to the manuscript are highlighted in red. We have included additional information in the manuscript in response to comments raised by the reviewers, and as a result, the sequential numerical order of the references has changed.

Response to the reviewer’s comment

Reviewer: 1

Comment #1

As the authors correctly identify the risks associated with MRSA, the work as a whole is interesting. You should definitely consider whether the layout of the work is logical enough and whether the text is fluent - some paragraphs do not flow smoothly one of the other. I am also missing a summary of bacteriophage therapy. Below are works that can help in the preparation of such a study:

https://www.mdpi.com/2076-2607/7/10/471

an interesting approach is also a combined bacteriophage-antibiotic therapy, little has been done here on the staphylococcal model, but there are other works available:

https://www.liebertpub.com/doi/abs/10.1089/mdr.2020.0083

Author’s response

Thank you for your valuable comment. As per the reviewer’s comment, the corresponding changes have been made and a summary about bacteriophage therapy and bacteriophage-antibiotic therapy has been included and the references were cited in the manuscript.

The authors sincerely record their heartfelt thanks to the editor and reviewers for your excellent effort in improving our manuscript according to the journal standard and scientific ethics. The revised manuscript has been resubmitted to your journal, and I request you to consider it favorably for publication in your esteemed journal. We look forward to your positive response.

Yours Sincerely

Dr. Mariappan Rajan

Reviewer 2 Report

Comments to the Author

I will begin first by congratulating the authors for their efforts to gather all this information and highlight it to the scientific community.

This paper needs minor revision and the following comments are suggested section-wise. Please incorporate these suggestions given as under

1-I have some remarks concerning the quality of the images,  for example, figure 3 page 12, I think it will be better to redraw the molecular structures and the quality will be excellent, the same for figure 1 page 3.

2-concerning the scientific content I do not know why the authors did not introduce the notion of the antimicrobial activity of surfactant molecules, a recent work of ''Tancer, Robert J., Kazim Baynes, and Gregory R. Wiedman . "Synergy among humimycins against methicillin‐resistant Staphylococcus aureus." Peptide Science 113.3 (2021): e24197'', in which the authors examine the activity of lamethicillin-based surfactants against Staphylococcus aureus. I suggest that the authors introduce the behavior of these types of molecules to further enhance their work.

3-For the Computational Analysis part, the authors have tried to give an approach to the usefulness of modeling to understand the behavior of antibiotics from the point of view of interactions, really the studies of molecular docking are very interesting but it should be noted that molecular docking is not always reliable, it must be linked by molecular dynamics in order to fully understand the behavior over time, which is very important to fully understand the reliability of these interactions, by studying different parameters such as the radius of gyration the RMSD, RMSF ...... that's why I suggest that the authors introduce this notion of molecular dynamics below the paragraph molecular docking to give clarity to the reader concerning the reliability of theoretical studies in bioinformatics, I suggest to the authors to read this paper "GIOIA, Dario, et al. Dynamic docking: a paradigm shift in computational drug discovery. Molecules, 2017, vol. 22, no 11, p. 2029" ,it will be useless to write this paragraph.

I have considered the following criteria to aid in my recommendation: 
A.      Conclusions adequately supported by the data: Yes
B.      Is the quality of figures acceptable: no
C.      Does the content of the manuscript justify the length: Yes 
D.      Appropriate referencing and up to date: Needs more recent references. 
E.      Is the level of English acceptable: yes
F.      Level of significance, novelty: Yes 

Author Response

Author’s Reply to Editor and Reviewers Comments

Journal                       : Antibiotics

Manuscript Title        : Recent Developments in Methicillin-Resistant Staphylococcus aureus (MRSA) Treatment: A Review

Manuscript Ref. No. : 1607808

Authors                      : Palanichamy Nandhini1, Pradeep Kumar2, Suresh Mickymaray3, Abdulaziz Al othaim3, Jayaprakash Somasundaram4 and Mariappan Rajan1,*

COMMENTS FROM THE AUTHOR:

The authors thank the editor and reviewers for the valuable comments and suggestions. As far as possible, we have answered the comments and suggestions given by the reviewers.

The changes to the manuscript are highlighted in red. We have included additional information in the manuscript in response to comments raised by the reviewers, and as a result, the sequential numerical order of the references has changed.

Response to the reviewer’s comment

Reviewer: 2

Comment #1

I have some remarks concerning the quality of the images, for example, figure 3 page 12, I think it will be better to redraw the molecular structures and the quality will be excellent, the same for figure 1 page 3.

Author’s response

Thank you for your valuable comment. As per the reviewer’s comment, figure-3 and figure-4 has been redrawn and the resolution of the figures has been improved.

Comment #2

Concerning the scientific content I do not know why the authors did not introduce the notion of the antimicrobial activity of surfactant molecules, a recent work of ''Tancer, Robert J., Kazim Baynes, and Gregory R. Wiedman. "Synergy among humimycins against methicillin‐resistant Staphylococcus aureus." Peptide Science 113.3 (2021): e24197'', in which the authors examine the activity of lamethicillin-based surfactants against Staphylococcus aureus. I suggest that the authors introduce the behavior of these types of molecules to further enhance their work.

Author’s response

Thank you for your valuable comment. As per the reviewer’s comment, a summary about antimicrobial activity of surfactant molecules have been included in the revised manuscript.

Comment #3

For the Computational Analysis part, the authors have tried to give an approach to the usefulness of modeling to understand the behavior of antibiotics from the point of view of interactions, really the studies of molecular docking are very interesting but it should be noted that molecular docking is not always reliable, it must be linked by molecular dynamics in order to fully understand the behavior over time, which is very important to fully understand the reliability of these interactions, by studying different parameters such as the radius of gyration the RMSD, RMSF ...... that's why I suggest that the authors introduce this notion of molecular dynamics below the paragraph molecular docking to give clarity to the reader concerning the reliability of theoretical studies in bioinformatics, I suggest to the authors to read this paper "GIOIA, Dario, et al. Dynamic docking: a paradigm shift in computational drug discovery. Molecules, 2017, vol. 22, no 11, p. 2029", it will be useless to write this paragraph.

Author’s response

Author thanks the reviewer suggestion for the betterment of manuscript. As per the reviewer's comment, a summary about molecular dynamics and its advantages have been included in the revised manuscript.

The authors sincerely record their heartfelt thanks to the editor and reviewers for your excellent effort in improving our manuscript according to the journal standard and scientific ethics. The revised manuscript has been resubmitted to your journal, and I request you to consider it favorably for publication in your esteemed journal. We look forward to your positive response.

Yours Sincerely

Dr. Mariappan Rajan

Reviewer 3 Report

The author's review is a well-organized description of recent developments in the treatment of MRSA. Beginning with the evolution of MRSA, the reader was reminded of the seriousness, and I was impressed that the current treatment and technical methods were analyzed clearly and accurately. The flow of the author's analysis, starting from the introduction, was natural without any difficulty in drawing conclusions, and it is judged that the contents are organically well connected.

However, I am curious about whether the images used do not have any copyright issues. Unless the excerpt is from a specific place, the authors are kindly asked that the figure be redrawn to improve readability. Most of the figures and chemical structures do not appear in normal aspect ratio. Also, please check the resolution.

Author Response

Author’s Reply to Editor and Reviewers Comments

Journal                       : Antibiotics

Manuscript Title        : Recent Developments in Methicillin-Resistant Staphylococcus aureus (MRSA) Treatment: A Review

Manuscript Ref. No. : 1607808

Authors                      : Palanichamy Nandhini1, Pradeep Kumar2, Suresh Mickymaray3, Abdulaziz Al othaim3, Jayaprakash Somasundaram4 and Mariappan Rajan1,*

COMMENTS FROM THE AUTHOR:

The authors thank the editor and reviewers for the valuable comments and suggestions. As far as possible, we have answered the comments and suggestions given by the reviewers.

The changes to the manuscript are highlighted in red. We have included additional information in the manuscript in response to comments raised by the reviewers, and as a result, the sequential numerical order of the references has changed.

Response to the reviewer’s comment

Reviewer: 3

Comment #1

The author's review is a well-organized description of recent developments in the treatment of MRSA. Beginning with the evolution of MRSA, the reader was reminded of the seriousness, and I was impressed that the current treatment and technical methods were analyzed clearly and accurately. The flow of the author's analysis, starting from the introduction, was natural without any difficulty in drawing conclusions, and it is judged that the contents are organically well connected.

However, I am curious about whether the images used do not have any copyright issues. Unless the excerpt is from a specific place, the authors are kindly asked that the figure be redrawn to improve readability. Most of the figures and chemical structures do not appear in normal aspect ratio. Also, please check the resolution.

Author’s response

Thank you for your valuable comment. As per the reviewer's comment, permission to copyright the figures have been purchased and the chemical structures have been redrawn also the resolution of the figures have been improved.

The authors sincerely record their heartfelt thanks to the editor and reviewers for your excellent effort in improving our manuscript according to the journal standard and scientific ethics. The revised manuscript has been resubmitted to your journal, and I request you to consider it favorably for publication in your esteemed journal. We look forward to your positive response.

Yours Sincerely

Dr. Mariappan Rajan